# Knowledge is not wisdom: Weight Balancing Mechanism for Local and Global Training in Federated Learning

## Abstract

Federated Learning (FL) is a unique approach that typically leverages client-side computing resources and data on edge devices. Data heterogeneity is a primary challenge that makes federated learning complex, and many studies have been conducted to address this issue. In previous studies, solutions were primarily focused on the client side, such as adjusting the weights of the local model or using proxy data from the aggregation server. However, we identified a problem where the global model becomes biased due to averaging the client's model, depending on the amount of the client's data or the extent of data sharing. Therefore, we introduce local and aggregation balancers for federated learning (FedBal), which respectively mediate the local training by class distribution and the weight aggregation by specific clients. We employ a local balancer to mitigate biases in favor of specific classes and an aggregation balancer to regulate biases toward certain clients. Remarkably, through experiments applying various existing methods with an aggregation balancer, we found that reflecting the models of marginalized clients more than those of clients with abundant data and classes can improve the accuracy of the global model by 2%–7%. FedBal, which combines two Balancers, exhibited an average accuracy improvement of 3%–4% compared to all other methods. This study raises several questions for further work to deepen our understanding of the role of the aggregation framework in FL.

## 1 Introduction

In the conventional deep neural network training paradigm, centralized algorithms are primarily used, and both computational resources and training datasets are integrated into a single server. However, with the advent of large-scale models and geographically distributed data, Federated Learning (FL), which utilizes multiple remote computing nodes, is gaining attention. The FedAvg algorithm (McMahan et al., 2017), which serves as the canonical optimization technique in FL, operates by maintaining a global model on the central server, synthesized by aggregating independently trained local models from multiple client nodes. While FedAvg demonstrates efficacy under conditions where client datasets are independently and identically distributed (IID) and client participation rates are elevated, it encounters challenges related to sluggish convergence rates under alternative conditions (Luo et al., 2021; Chen et al., 2023; Vahidian et al., 2023). Specifically, as the number of local iterations increases for training efficiency and reduces network capacity, each client's model becomes increasingly biased towards its own data, which is the so-called "client drift"(Zhao et al., 2018). Furthermore, FL systems are susceptible to "canceling out" effects in classifiers due to variations in label distribution (label skewness) and data quantity among clients(Liao et al., 2023).

Several studies primarily focused on enhancing local learning algorithms through well-designed regularization and aggregation via FedAvg (or uniform averaging) to mitigate data heterogeneity (Li et al., 2020; Karimireddy et al., 2020; Wang et al., 2020b). However, we identified a problem where the global model becomes biased due to averaging the client's model, depending on the amount of the client's data or the extent of data sharing (see Figure 1a). This issue requires adjustments not only in local regularization but also at the aggregation, where the states of each client can be compared. From the perspective of model aggregation, various methods have been proposed to expedite convergence or to address non-iid issues. This issue requires adjustments not only in local regular-

ization but also at the aggregation level, where the states of each client can be compared. From the perspective of model aggregation, various methods have been proposed to expedite convergence or address non-IID issues. Still, these methods (Zhao et al., 2018; Yoon et al., 2021) either pose data exposure risks when utilizing proxy or augmented data or entail complex structures as they adopt training paradigms (Wang et al., 2020a; Tang et al., 2021; Wu & Wang, 2022; Fraboni et al., 2021).

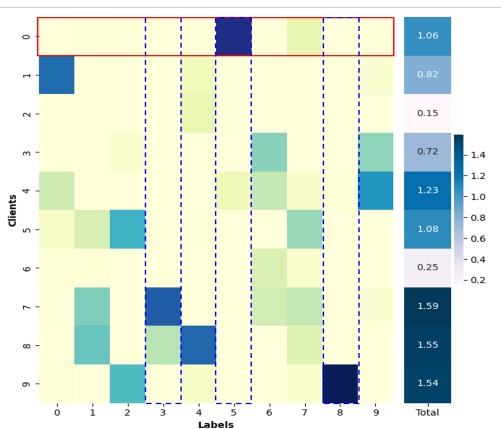

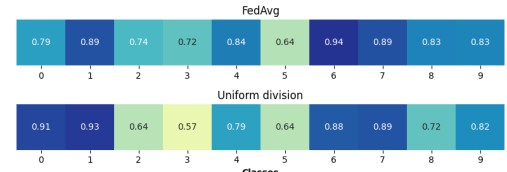

(a) Example of data distribution in the CIFAR-10 dataset on random distribution (details on Appendix A.1). The X- and Y-axes represent the data label and client ID, respectively. The darker the color, the more data is available to the client. On the right side of the figure, 'Total' refers to the total data of each client. The red solid box represents the labels that 'client 0' holds, and the blue dashed box represents inaccurate labels in aggregate with equal division.

(b) Class-wise accuracy when performing FL through FedAvg and uniform division, based on the data distribution in 1a. Darker color means the more accurate class. The horizontal axis represents the classes of data.

Figure 1: When employing FedAvg, the global model exhibits bias towards the labels (1, 6, 7, etc.) held by a specific client. Conversely, with uniform division, labels (0, 1, 6, 7, etc.) that are more widely shared among clients yield higher accuracy. Label 5 is marginalized in both scenarios, leading to reduced accuracy in each case.

In this work, we propose a local and aggregation balancer (**FedBal**). Client drift negatively affects the global model's accuracy as each client relies on its own data for training. To mitigate "client drift," we implement a local balancer that adjusts the loss of classes each client has, ensuring the learning is not biased towards specific classes. Additionally, we devise an aggregation balancer at the global aggregation stage—the only point where the weight distribution of all clients can be observed—to resolve the "canceling out" effect by automatically determining the aggregation ratio through similarity calculations with the global model. While adapting the aggregation balancer, we were confronted with a naive philosophical question: *Is a client with extensive knowledge always wise?* Surprisingly, we found that enhancing the overall model's performance is not achieved by reflecting more weight from clients that have undergone extensive learning and thus differ significantly from the global model. Instead, reflecting more weight on clients with less variation reduces overfitting in the global model. Finally, the aggregation balancer has been proven to be easily applicable and mostly effective with existing methods.

**Our Contribution.** The proposed framework is subjected to empirical evaluation using benchmark datasets that exhibit varying degrees of data heterogeneity. This assessment substantiates the framework's efficacy in achieving superior model performance. When the aggregation balancer was applied to existing models, there was an average performance improvement of 2%–7% depending on the level of non-IIDness. FedBal, which combines two Balancers, exhibited an average accuracy improvement of 3%–4% compared to all other methods. Our main contributions are summarized as follows:

- Introduce a balancing mechanism to prevent a biased global model towards a client or class.

- We empirically demonstrate that giving more attention to overlooked clients, instead of those with lots of varied data, at the aggregation stage helps prevent overfitting.

- In the context of FL, we have found that it is more appropriate to determine the aggregation ratio concerning the classifier than the feature extraction layer.

## 2 PROBLEM SETTING AND NOTATIONS

Given $M$ clients, the objective of FL is to ascertain a global model $\theta$ that alleviates the average losses across all clients, as illustrated below:

$$\underset{\theta \in \mathbb{R}^d}{\arg\min}\, L(\theta) := \sum_{i=1}^{M} p_i L_i(\theta) \quad \text{and} \quad L_i(\theta) = \mathbb{E}_{(x,y) \sim D_i}[L_i(x, y; \theta)] \tag{1}$$

The function $L_i$ measures the average loss of a model with parameters $\theta$ on the $i^{th}$ client's training data $D_i$, and $\sum_{i=1}^{M} p_i = 1$ which is the weight given to client $i$. Typically, in vanilla FedAvg, $p_i$ is determined to be proportional to the number of samples from the client $i$. On the other hand, depending on the implementation, it can be assigned uniformly. The objective is to identify a model that adequately fits all clients' data on a weighted average. It is imperative to note that clients may exhibit heterogeneous data distributions, and any exchanges of training data are explicitly forbidden due to prevailing privacy concerns. Our objective is to develop a Local Balancer, $L_i(\theta)$, that addresses class imbalance and an Aggregation Balancer that finds the optimal $p_i$ capable of minimizing $L(\theta)$.

In a general FL framework, the server simply aggregates all the participating client models to obtain the global model. Specifically, in the $t^{th}$ communication round, a central server first sends a global model $\theta^{t-1}$ to each of the clients. Each client sets their initial model $\theta^t$ to $\theta^{t-1}$, performs $K$ steps of the gradient descent optimization to minimize its local loss, and then returns the resultant model to the central server. The global model for the subsequent round is derived by averaging all the local models from participating clients in the current round of communication.

### 2.1 PROBLEM OF FEDAVG

FedAvg can reflect client weights in the global model either based on the size of the data or uniformly, depending on the implementation and the setting of $p_i$. When $p_i$ is determined by the data size, as depicted in Figure A.1, Clients 7, 8, and 9 collectively hold a 50% stake in the global model. Consequently, the accuracy(see 'FedAvg' in Figure 1b) is generally high for the classes they possess, but it is relatively low for un-holding classes(e.g., 0, 5, 9). On the other hand, when distributed uniformly, the reflection ratio $p_i$ is determined by how many clients share a particular label. This leads to differences in accuracy depending on the degree of data sharing, as in 'uniform division' from Figure 1b. The class-wise accuracy of clients sharing less data (blue dash box in Figure A.1) is lower compared to the accuracy of other classes. Label 5, which is marginalized in both scenarios, has low accuracy in both cases.

For optimal averaging, it is imperative that clients engage in proactive measures to address the non-IID phenomena inherent in their specific contexts. Given that client models are also subjected to learning within imbalanced states, it becomes intricate to integrate information pertaining to underrepresented labels into the global model. Furthermore, meticulous examination of each client's distinctive attributes is essential, ensuring the seamless incorporation of diverse information into the global model without inducing conflicts. To address the aforementioned challenges, our study employs a Local Balancer and introduces a Aggregation Balancer, strategically aligning with the respective needs of each scenario.

## 3 RELATED WORK

FL has emerged as a prominent paradigm, allowing model training across multiple decentralized devices (or clients) holding local data samples, thus avoiding exchanging them. This approach is particularly beneficial for preserving privacy and reducing the need to transfer large amounts of data

to a central location. However, one of the inherent challenges in FL is dealing with non-IID data, where the data distribution varies significantly across clients. Thus far, methods have been applied to the client or the global server to address these issues in FL research.

**Client side applied methods.** A spectrum of research concentrates on refining local learning algorithms, designed to regularize model weights and implementing local bias correction. FedProx (Li et al., 2020) incorporates a proximal term into the local training objective to maintain the consistency of the updated parameter with the original model. Similarly, SCAFFOLD (Karimireddy et al., 2020) employs cross-client variance reduction in local updates. FedNova (Wang et al., 2020b) a normalized gradient averaging method that eliminates objective inconsistency while preserving fast error convergence. This method represents a significant advancement in tackling the objective inconsistency problem in heterogeneous federated optimization. Recently, methods that employ knowledge distillation to regularize client weights have emerged (Kim et al., 2022). Besides, methods such as class-balanced data re-sampling and loss re-weighting have proven efficacious in enhancing training performance in scenarios where clients possess imbalanced local data (Hsu et al., 2020; Wang et al., 2021; Xu et al., 2023). However, most existing methods predominantly constrain weights based on the global model. If the global model is biased towards specific clients or classes, it inadvertently hinders training diverse class distributions. The proposed local balancer does not compare with the global model; instead, it adjusts the logits based on each client's class distribution.

**Server side applied methods.** From the server side, studies have been conducted using methods that select clients advantageous for training or utilize public data to enhance the aggregation accuracy. (Wang et al., 2020a) employed reinforcement learning to select clients in a manner that promotes the enhancement of validation accuracy while imposing penalties on the utilization of communication rounds. FedGP (Tang et al., 2021) boosts the convergence rate of FL using loss correlations between the clients. (Wu & Wang, 2022) and (Fraboni et al., 2021) proposed dynamically changing the probability for each client to be selected. Additionally, (Wang et al., 2021) used local and global imbalances similarly to our approach. However, the most significant difference is that they share the class information they hold during the global aggregation phase. Thus, the proposed Aggregation Balancer does not directly observe the data; instead, it only compares the weights to dynamically change the participation rate of the clients. Moreover, our method can be applied without the need for additional training.

## 4    PROPOSED METHOD: FEDBAL

We propose balancers that operate in local training and aggregation phases in FL framework. The local balancer alleviates weight divergence by utilizing the class label counter, for which (Li et al., 2022a) gives us direct motivations. On the other hand, the aggregation balancer is a mechanism that reflects more on marginalized clients by comparing each client's and the previous global model classifier weight. The aggregation balancer can be applied as a substitute for FedAvg in different ways as well.

### 4.1    LOCAL BALANCER

In FL, due to the non-IID dataset, insufficient or absent labels are occurrences at each client. In such cases, they are trained unevenly at the local level. When the data is imbalanced, the softmax works for a given sample as follows:

$$L_{\text{softmax}}(x) = -\log \frac{e^{z_y}}{\sum_j e^{z_j}} \tag{2}$$

The gradient of $L_{\text{softmax}}$ w.r.t. $z_i$ is:

$$\frac{\partial L_{\text{softmax}}}{\partial z_i} = \begin{cases} \hat{p}_i - 1, & i = y \\ \hat{p}_i, & i \neq y \end{cases} \tag{3}$$

where $\hat{p}_i = \frac{e^{z_i}}{\sum_j e^{z_j}}$. $z_i$ represents the predicted logit of class $i$, therefore $z_y$ indicates the target logit and $z_i (i \neq y)$ is the non-target logit. Gradients corresponding to the target class are negative during

backward propagation, whereas those for non-target classes are positive. Consequently, the training samples impose penalties on the weights (where $i \neq y$), of the non-target classes by $\hat{p}_i$. Therefore, more rewards need to be given to the non-target class to achieve training balance. Additionally, as mentioned in (He et al., 2018), the larger the representation norm, the more it implies that there are more learned representations. This is maximized to induce both representation and logit to be well learned in local training. Since the goal of our local balancer is to alleviate client drifts, we directly apply the method used in (Li et al., 2022a) as below.

$$L_{\text{LB}} = -\frac{1}{N} \sum_i \log \frac{e^{z_{y_i}^b}}{\sum_j e^{z_j^b}} \tag{4}$$

where the $z_j^b$ represents the balanced logit of class $j$, and $N$ is number of samples in batch. Detailed explanations for formulas are in the Appendix B.

## 4.2 AGGREGATION BALANCER

The Aggregation Balancer was employed due to the appearance of clients that monopolized specific labels. These clients are being marginalized by the global model. Here, a naive philosophical question occurred to us: *Is a client with extensive knowledge always wise?* From a societal perspective, when making crucial decisions, it may be rational to adhere to the opinions of experts(the majority) with diverse and in-depth knowledge. However, it is wiser also to take the views of the marginalized minority. We also figured out that the same phenomenon is occurring in FL(see in Section 5.2). Incorporating too much weight from clients who possess abundant data and labels can quickly lead to overfitting, and as the global iteration increases, the model collapses. However, including more clients who hold unique data and whose local training is insufficient due to having less data with a rich distribution enhances the overall performance of the global model. From this motivation, we devised the aggregation balancer to reflect the knowledge of the minority while still utilizing significant clients to create a wise global model. We aim to determine the proper $p_i$ in equation 1, allowing marginalized clients more opportunities to be reflected in the global model.

To achieve the goal, the aggregation balancer waits until all client models are transmitted to achieve the goal. Once all the client models have arrived, it calculates the importance score as follows:

$$v'_{i,\text{score}} = \text{cos\_similarity}(\theta_{\text{cls}}^{t-1}, \theta_{i,\text{cls}}^t) = \frac{\theta_{\text{cls}}^{t-1} \cdot \theta_{i,\text{cls}}^t}{\|\theta_{\text{cls}}^{t-1}\|_2 \cdot \|\theta_{i,\text{cls}}^t\|_2}, \quad \text{for all} \quad i \in \{1, \ldots, M\} \tag{5}$$

where the $\theta_{\text{cls}}$ is the classifier parameter. The importance score of each client $v_{i,\text{score}}$ is measured by cosine similarity between the global model $\theta^{t-1}$ before local learning and $\theta_i^t$ which is a trained model using the local dataset. And then we append $v'_{i,\text{score}}$ in a one vector space:

$$\mathbf{v}'_{\text{score}} := [v'_{1,\text{score}}, \ldots, v'_{M,\text{score}}] \tag{6}$$

The reason for comparing cosine similarity only to the classifier, not the entire model, is 1) because the classifier gets strong feedback from true labels and 2) because as seen in Appendix C the representation layer does not change significantly as the global model updates. In other words, the process of comparing classifiers is both efficient and effective. While norms (e.g., l2-norm) can be used to measure similarity, norms consider both the angle and the magnitude between two vectors, resulting in a value ultimately proportional to the size of the data. This makes it challenging to select clients holding unique classes. This is because clients possessing unique classes may hold relatively less data.

However, if $\mathbf{v}'_{\text{score}}$ has a large deviation due to the dissimilar models, this also adversely affects the global model. Therefore, to prevent dissimilar client models (which means that update more weights), clipping is performed being overly reflected in the global model. Let $\bar{v}$ represents the mean of $\mathbf{v}'_{\text{score}}$ and $\sigma$ is the standard deviation. Define a function $f : \mathbb{R} \to \mathbb{R}$ that perform clipping by:

$$\mathbf{v}^{\text{clip}} = f(\mathbf{v}'_{\text{score}}), \quad f(\cdot) = \begin{cases} v_i & \text{if } v_i \geq T \\ T & \text{if } v_i < T \end{cases} \tag{7}$$

Here, $T = \bar{v} - \beta * \sigma$ which is threshold. $\beta > 0$ controls the outlier level. Also, the $\beta$ value can be used as a temperature parameter (as in the softmax function) to control the smoothness of the probability distribution. In our experiment, $\beta$ is set to 3 in constant. The smaller value of $\mathbf{v}^{\text{clip}}$ mplies a dissimilar client model to the previous global model. To answer the initial question, *"Is a client with extensive knowledge always wise?"*, we apply the softmax function directly to the $\mathbf{v}^{\text{clip}}$ distribution to create a probability distribution. Therefore, the final form of the importance score is as follows:

$$\mathbf{v}_{\text{score}} = \frac{e^{v_i^{\text{clip}}}}{\sum_j e^{v_j^{\text{clip}}}} \tag{8}$$

Now, $\mathbf{v}_{\text{score}}$ is a probability function that can replace $p_i$ in Equation 1. Our final **FedBal** is expressed as:

$$\underset{\theta \in \mathbb{R}^d}{\arg\min} \, L(\theta) := \sum_{i=1}^{M} \mathbf{v}_{\text{score}} L_{\text{LB}}^i(\theta) \quad \text{and} \quad L_{\text{LB}}^i(\theta) = \mathbb{E}_{(x,y) \sim D_i}[L_{\text{LB}}^i(\theta^t)(x, y; \theta)] \tag{9}$$

Further implementation details and reasoning for hyper-parameters can be found in Appendix C.

## 5 EXPERIMENT AND EVALUATION

### 5.1 EXPERIMENTAL SETUP

**Datasets and Partitioning.** We conducted experiments using CIFAR-10 and CIFAR-100 with ConvNet that introduced from (Shin et al., 2022). Most of the experiments, including hyper-parameter tuning, were performed using CIFAR-10, while CIFAR-100 was utilized to verify the robustness of the method. At this time, the hyper-parameter values were fixed to those tuned with the CIFAR-10 dataset. In this study, only cross-silo environments were considered, and it was tested in a scenario where only 4 clients participated in each global round(20% of a total of 20 clients). The datasets were pre-distributed to all clients according to the Dirichlet alpha ratio, and the clients to participate in the global round were randomly selected. Every client has a unique data set with no duplicate data. The models of all participating clients are weighted, averaged, and then applied to the global model.

All datasets were composed with Dirichlet distribution values $\alpha \in [0.05, 0.1, 0.5, 1]$ as in (Li et al., 2022b). A smaller Dirichlet $\alpha$ value implies a more non-iid setting (see Appendix A.1).

**Models and Implementation.** Our experiments primarily used a 3-layered ConvNet with the same structure as in (Oh et al., 2021; Shin et al., 2022). Similarly, to verify the model robustness, ResNet-18 is used. The detailed model architecture is addressed in the Appendix A.2. For the learning setup, we applied local epoch $K$ 10 times, and Global round $T$ was carried out 130 times. We utilized SGD for optimization, with momentum and weight decay set at 0.9 and 1e-4, respectively. We employed a local learning rate of 0.01 and a global learning rate of 1.0, and learning rate decay was utilized, as in (Shin et al., 2022). In conjunction with PyTorch, we have implemented the FL framework utilizing Ray(Moritz et al., 2018), which is a framework designed for distributed learning. All experimental evaluations were executed utilizing two Nvidia 3090 GPUs.

Our experiments primarily used a 3-layered ConvNet with the same structure as in (Oh et al., 2021; Shin et al., 2022). Similarly, to verify the model's robustness, ResNet-18 is used. The detailed model architecture is addressed in Appendix A.2. For the learning setup, we applied local epoch K 10 times, and global round T was carried out 130 times. We utilized SGD for optimization, with momentum and weight decay set at 0.9 and 1e-4, respectively. We employed a local learning rate of 0.01 and a global learning rate of 1.0, and learning rate decay was utilized, as in (Shin et al., 2022)). In conjunction with PyTorch, we have implemented the FL framework utilizing Ray (Moritz et al.,

2018), which is a framework designed for distributed learning. All experimental evaluations were executed utilizing two Nvidia 3090 GPUs.

## 5.2 IS A CLIENT WITH EXTENSIVE KNOWLEDGE ALWAYS WISE?

To answer this question, additional experiments were conducted. The experiments were conducted using the CIFAR-10 dataset and the ConvNet model, with FedBal serving as the baselines. And all other conditions were the same as mentioned in Section 5.1, and an inversely proportional term, which is $\exp(\mathbf{v}_{score})$, was added in equation 8. As can be seen from the experimental results(Table 1), it was confirmed that giving more weight to marginalized clients aids in overall performance improvement. There was an elevating effect in accuracy and a small deviation. The experiment was conducted a total of three times.

Table 1: Accuracy evaluation for inversely proportional term.

| Type | With $\exp$ | Without $\exp$ (Original) |
|---|---|---|
| Accuracy | 63.24 ($\pm 1.37$) | **64.50** ($\pm 1.06$) |

## 5.3 EXPERIMENT RESULTS

The results of this experiment were obtained by performing 10 local iterations and 130 global iterations on CIFAR-10 and CIFAR-100 with ConvNet (see the details in Appendix A.2). While distributing data with Dirichlet alpha, there is a concern that there will be variations in difficulty depending on the class label. Therefore, in this study, all methods used the same data distribution, and to prevent the convergence speed from varying depending on the model's initial value, the same initial value was used as well. However, the clients participating in the training were sampled randomly.

**Results on Local Balancer.** The local balancer aims to mitigate client drifts within local learning. Table 2 provides a summary according to different Dirichlet alpha values. As can be seen from the results, the Local Balancer operates within the classes it possesses. Hence, it exhibits good performance in IID situations. It even outperforms Scaffold by more than 5% with $\alpha = 1.0$. However, in extreme non-IID cases (e.g., $\alpha$ with 0.05), its performance falls behind FedAvg. This phenomenon is also observed for FedNova. Consequently, the employment of the local balancer in isolation is unfeasible. In the presence of Non-IID conditions, a mechanism is indispensable to mitigate the inherent vulnerabilities of the local balancer. The aggregation balancer we suggest can efficaciously serve as a surrogate to fulfill this imperative role.

Table 2: Top-1 accuracy of the local balancer depends on data skewness, which has more advantages in IID situations on both CIFAR-10 and CIFAR-100. The bold font represents the best result.

| Datasets | CIFAR-10 | | | | CIFAR-100 | | | |
|---|---|---|---|---|---|---|---|---|
| Skewness | $\alpha$=0.05 | $\alpha$=0.1 | $\alpha$=0.5 | $\alpha$=1.0 | $\alpha$=0.05 | $\alpha$=0.1 | $\alpha$=0.5 | $\alpha$=1.0 |
| FedAvg | 54.81 | 59.90 | **66.01** | 67.13 | 33.40 | 35.27 | 33.35 | 34.04 |
| FedProx | **57.90** | 62.34 | 65.93 | 67.26 | 34.18 | **36.23** | 32.62 | 34.49 |
| Scaffold | 55.45 | 62.99 | 64.76 | 63.56 | 30.20 | 32.33 | 33.07 | 31.79 |
| FedNova | 52.73 | **64.35** | 65.61 | 67.78 | 32.93 | 35.75 | 34.15 | 34.33 |
| LocalBalancer (Ours) | 52.41 | 62.34 | 65.09 | **68.78** | **34.54** | 35.43 | **35.52** | **35.13** |

**Adapting Aggregation Balancer for other methods.** The aggregation balancer needs to reflect the marginalized classes in non-IID data distributions more in the global model. According to Table 3, the aggregation balancer is more effective as it becomes more non-IID and effective in all methods when $\alpha = 0.05$. In particular, significant improvements were observed when utilizing the local balancer. This can be seen as the aggregation balancer supplementing the insufficient information as the local balancer learns the nonexistent label distribution.

Depending on the degree of Non-IID, excluding when $\alpha = 0.1$, there was an accuracy improvement exceeding 1% in many cases from the CIFAR-100 test. The effect is not as pronounced as in CIFAR-

10, but considering that CIFAR-100 is currently a dataset experiencing difficulties in FL, it can be regarded as a notable effect.

Table 3: Top-1 accuracy for adapting the aggregation balancer in CIFAR-10. The numbers inside the parentheses represent the accuracy differences when the aggregation balancer was applied.

| Dataset | CIFAR-10 | | | |
|---|---|---|---|---|
| Skewness | $\alpha$=0.05 | $\alpha$=0.1 | $\alpha$=0.5 | $\alpha$=1.0 |
| FedAvg | 54.81 | 59.90 | 66.01 | 67.13 |
| FedAvg w/ AB | 58.62 (+3.81) | 63.11 (+3.21) | 68.22 (+2.21) | 67.05 (-0.08) |
| FedProx | 57.90 | 62.34 | 65.93 | 67.26 |
| FedProx w/ AB | 61.43 (+3.53) | 63.63 (+1.29) | 66.33 (+0.4) | 67.90 (+0.64) |
| Scaffold | 55.45 | 62.99 | 64.76 | 63.56 |
| Scaffold w/ AB | 57.54 (+2.09) | 62.79 (-0.20) | 65.17 (+0.41) | 66.65 (+3.09) |
| FedNova | 52.73 | 64.35 | 65.61 | 67.78 |
| FedNova w/ AB | 59.30 (+6.57) | 63.27 (-1.08) | 67.54 (+1.93) | 69.06 (+1.28) |
| LocalBalancer (Ours) | 52.41 | 62.34 | 65.09 | 68.78 |
| FedBal (Ours) | 59.70 (+7.29) | 63.95 (+1.61) | 68.22 (+3.13) | 69.70 (+0.92) |

Table 4: CIFAR-100 result for adapting Aggregation Balancer(AB). The values inside the parentheses represent the difference when the AB is applied.

| Dataset | CIFAR-100 | | | |
|---|---|---|---|---|
| Methods | $\alpha$=0.05 | $\alpha$=0.1 | $\alpha$=0.5 | $\alpha$=1.0 |
| FedAvg | 33.40 | 35.27 | 33.35 | 34.04 |
| FedAvg w/ AB | 33.21 (-0.19) | 34.94 (-0.33) | 34.67 (+1.32) | 35.09 (+1.05) |
| FedProx | 34.18 | 36.23 | 32.62 | 34.49 |
| FedProx w/ AB | 34.62 (+0.44) | 35.14 (-1.13) | 35.12 (+2.50) | 35.49 (+1.00) |
| Scaffold | 30.20 | 32.33 | 33.07 | 31.79 |
| Scaffold w/ AB | 30.52 (+0.32) | 31.20 (-0.65) | 31.21 (-1.86) | 31.91 (+0.12) |
| FedNova | 32.93 | 35.75 | 34.15 | 34.33 |
| FedNova w/ AB | 35.14 (+2.21) | 35.10 (-1.41) | 34.67 (+0.52) | 35.90 (+1.57) |
| LocalBalancer (Ours) | 34.54 | 35.43 | 35.52 | 35.13 |
| FedBal (Ours) | 34.90 (+0.36) | 34.02 (-1.41) | 34.96 (-0.56) | 36.18 (+1.05) |

**Effectiveness of Aggregation Balancer** According to this study, the aggregation balancer reflects more of the weights of marginalized clients in the early stages, but it allocates weights in equal proportions in the later stages (see more details in Appendix C.3). However, for clients that have undergone extensive learning, there is a significant change in magnitude, even if there is little change in the gradient angle. Therefore, ultimately, in the later stages of learning, clients with more data play a more significant role in forming the global model. We discerned the significance of the role of the aggregation balancer, which reflects wiseness, especially in the initial stages where the differences in training due to the amount of data are pronounced.

**Comparison between FedBal and other methods.** Table 5 represents the difference in accuracy when FedBal was applied. Except for the situation where $\alpha = 0.1$, it showed better results than the existing methods in all cases. FedBal can expect accuracy improvement in Non-IID situations due to the Aggregation Balancer, and in IID situations, its weaknesses can be compensated for by the Local Balancer. Specifically, FedBal outperformed FedNova by 6.97% when $\alpha = 0.05$. Even compared to the robust baseline, FedAvg, it exhibited approximately 5% greater efficacy. While the performance of other methods varied significantly depending on the degree of non-IID, the difference in the performance of FedBal was not substantial. Moreover, even in IID situations, such as when $\alpha = 1.0$, it exhibited a performance that was 6.14% better than Scaffold, and compared to other methods, there was an approximately 2% improvement in performance.

In both CIFAR-10 and CIFAR-100 cases, the accuracy is relatively low when $\alpha = 0.1$. We hypothesize that this is due to the occurrence of a balancer bottleneck. A balance bottleneck is a section where the local and aggregation balancers conflict, offsetting each other's advantages.

Table 5: Top-1 accuracy comparison on CIFAR-10 with ConvNet. The number inside the parentheses represents the accuracy differences between FedBal and previous methods. The bold font represents the best results.

| Dataset | CIFAR-10 | | | |
|---|---|---|---|---|
| Skewness | $\alpha$=0.05 | $\alpha$=0.1 | $\alpha$=0.5 | $\alpha$=1.0 |
| FedAvg | 54.81 (-4.89) | 59.90 (-4.05) | 66.01 (-2.21) | 67.13 (-2.57) |
| FedProx | 57.90 (-1.80) | 62.34 (-1.61) | 65.93 (-2.29) | 67.26 (-2.44) |
| Scaffold | 55.45 (-4.25) | 62.99 (-0.96) | 64.76 (-3.46) | 63.56 (-6.14) |
| FedNova | 52.73 (-6.97) | **64.35 (+0.40)** | 65.61 (-2.61) | 67.78 (-1.92) |
| FedBal (Ours) | **59.70** | 63.95 | **68.22** | **69.70** |

Table 6: Top-1 accuracy comparison on CIFAR-100. The number inside the parentheses represents the accuracy differences between FedBal and previous methods. The bold font represents the best results.

| Dataset | Cifar-100 | | | |
|---|---|---|---|---|
| Skewness | $\alpha$=0.05 | $\alpha$=0.1 | $\alpha$=0.5 | $\alpha$=1.0 |
| FedAvg | 33.40 (-1.50) | 35.27 (+1.25) | 33.35 (-1.61) | 34.04 (-2.14) |
| FedProx | 34.18 (-0.72) | **36.23 (+2.21)** | 32.62 (-2.34) | 34.49 (-1.69) |
| Scaffold | 30.20 (-4.70) | 32.33 (-1.69) | 33.07 (-1.89) | 31.79 (-4.39) |
| FedNova | 32.93 (-1.97) | 35.75 (+1.73) | 34.15 (-0.81) | 34.33 (-1.85) |
| FedBal (Ours) | **34.90** | 34.02 | **34.96** | **36.18** |

## 5.4 DISCUSSION

In ResNet, the aggregation balancer was ineffective in all methods (refer to Appendix D). This is presumed to be because, unlike ConvNet, ResNet learns a lot of information in the feature extraction layer. Nevertheless, our approach calculates similarity by considering only the classifier weights, disregarding the information inherent to the feature extraction layer. Thus, in ResNet, adjusting the representation is more effective than adjusting the classifier. In subsequent research, we intend to further observe the learning characteristics of ResNet and devise strategies suitable for deeper models.

## 6 CONCLUSION AND FUTURE WORK

This research endeavored to mitigate the inherent biases amongst classes and clients induced by non-IID distributions in the FL framework. A local balancer, introduced during the local training stage, enables the correction of underrepresented classes. Concurrently, an aggregation balancer constructs a global model in the aggregation stage by setting ratios based on the observed differences between client model distributions. This methodology was substantiated using the CIFAR-10 dataset and a ConvNet model. Future research will focus on developing a robust local balancer capable of operating effectively in extreme non-IID environments and an adaptable aggregation balancer that is modifiable under data distribution and model status.

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

# Appendix

## A  DETAILS OF EXPERIMENTAL SETUP

### A.1  DATASETS AND DATA PARTITIONING

This study utilized the CIFAR-10 and CIFAR-100 datasets. The CIFAR-10 dataset comprises 60,000 images, each of $32 \times 32$ size, categorized into 10 distinct classes, each containing 6,000 images. A total of 5,000 images were allocated as training data and 1,000 as test data. Similarly, the CIFAR-100 dataset encompasses 60,000 images, each of $32 \times 32$ size, segregated into 100 classes, each harboring 600 images. These 100 classes were further grouped into 20 superclasses that were not used in our experiments. Furthermore, 500 images were designated as training data and 100 as test data in this dataset.

The class was partitioned before being allocated to the client according to the intensity of the Dirichlet distribution. The data assigned to each client were distinct, with no intersections, and the aggregate ratio of all distributed class data was 1. However, the total quantity of data that each client possessed was variable. Figure 2 provides an exemplar of data distributed in alignment with the alpha value.

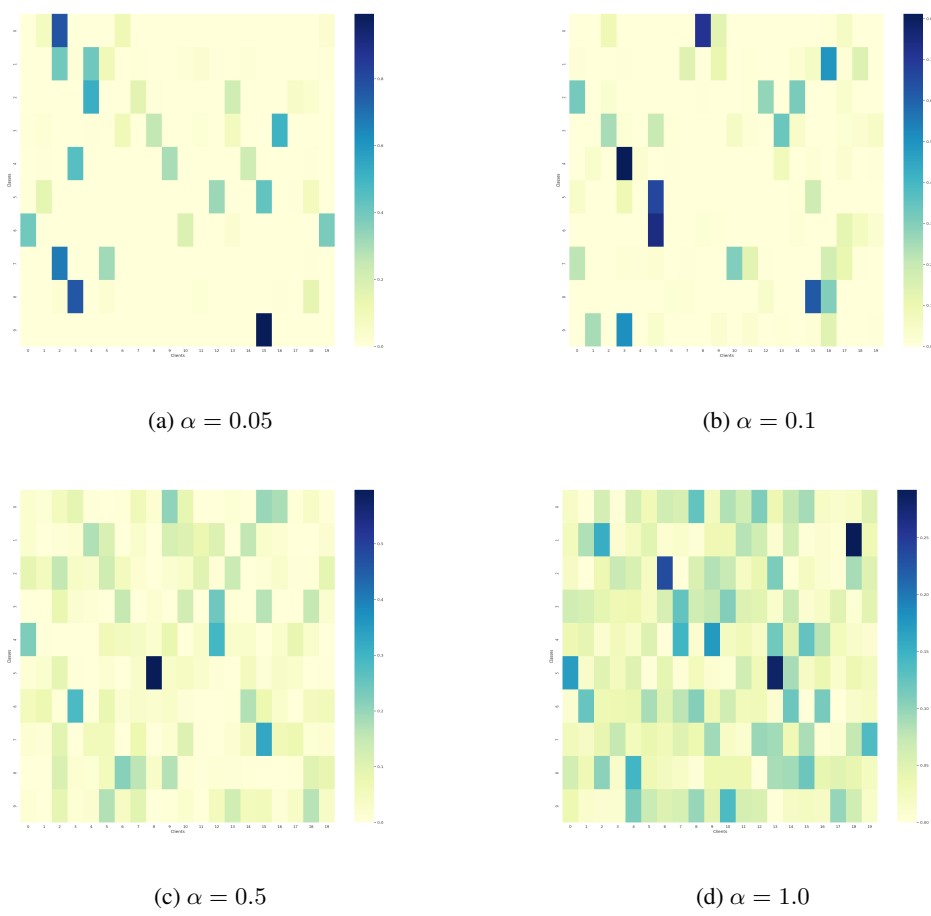

(a) $\alpha = 0.05$                    (b) $\alpha = 0.1$

(c) $\alpha = 0.5$                    (d) $\alpha = 1.0$

Figure 2: Example of data distribution according to the Dirichlet distribution. The abscissa denotes the number of clients, which is predetermined at 20. The ordinate symbolizes the label, wherein a darker shade signifies a higher numerical value. The degree of non-IID increases concomitantly with the number of classes not possessed by the client.

## A.2 Model Architecture

The CNN structure we utilized consists of three consecutive convolution layers, one max pooling layer, and directly connects to one fully connected layer. The shape of the convolution layer is represented as $(C_{in}, C_{out}, C_{kernel}, C_{kernel})$, and $(C_{in}, C_{out})$ for the fully connected layer. All non-linear activation functions employed ReLu. Table 7 provides the detailed information.

Table 7: Detailed information of the CNN architecture used in our experiments.

| Parameter | Shape | Layer hyper-parameter |
|---|---|---|
| conv1.weight | (3, 64, 3, 3) | stride=1, padding=1 |
| conv1.bias | (64) | N/A |
| conv2.weight | (64, 64, 3, 3) | stride=1, padding=1 |
| conv2.bias | (64) | N/A |
| conv3.weight | (64, 128, 3, 3) | stride=1, padding=1 |
| conv3.bias | (128) | N/A |
| MaxPool | N/A | stride=2, padding=1 |
| fc1.weight | (36992, number of classes) | N/A |
| fc1.bias | (number of classes) | N/A |

## B Formulation of Local Balancer

According to the (Li et al., 2022a), increasing the influence of marginalized classes is crucial for training while enhancing the corresponding representation norm. An additional constraint item can be incorporated into the original cross-entropy loss to induce a substantial feature norm:

$$L'_{LB} = -\log \frac{e_y^z}{\sum_i e_i^z} + \alpha \frac{\lambda_y}{\|\mathbf{f}\|} \tag{10}$$

where $\alpha$ serves as the parameter employed to modulate the strength of the constraint, and $\lambda_y$ governs the intensity of the stimulus directed towards different classes. To prevent the norm from being suppressed in underrepresented classes, $\lambda_y$ is assigned to be negatively correlated with the number of instances per class:

$$L'_{LB} = -\log \frac{e^{z_y}}{\sum_j e^{z_j}} + \log e^{\frac{\lambda_y}{\|\mathbf{f}\|}}$$
$$= -\log \frac{e^{z_y - \frac{\lambda_y}{\|\mathbf{f}\|}}}{\sum_j e^{z_j}} \tag{11}$$

As the sum of the probabilities of all classes obtained by Eq. 11 is not equal to 1. (Li et al., 2022a) additionally adjusted the logit to ascertain that the cumulative predicted probabilities across all classes equate to 1:

$$z_j^b = z_j - \alpha(k) \frac{\lambda_j}{\|\mathbf{f}\|} \tag{12}$$

where the $z_j^b$ represents the balanced logit of class $j$. Additionally, $\alpha(k)$ is the learning strategy that controls the regularization intensity. For this experiment, we set $\alpha(k)$ as:

$$\alpha(k) = \left(\frac{k}{K}\right)^2 \tag{13}$$

where the $k$ is current local iteration and $K$ represents the total number of local iteration. Initially, the model prioritizes the original loss (e.g., cross-entropy loss) in early training and gradually increases the intensity of regularization. Finally, Local Balancer is expressed as in Eq. 4.

## C   EVALUATION OF AGGREGATION BALANCER

### C.1   WHY ONLY CALCULATES THE CLASSIFIER, NOT THE WHOLE MODEL?

The classifier is a layer directly associated with labels. Therefore, it is most sensitive to label distribution. To verify this, we trained a model with a data distribution with a Dirichlet alpha value of 0.1 in FedAvg. For simplicity, just 30 global rounds were executed. We plotted the cosine similarity between the previous global model and the aggregated model of each layer as a heat map. Indeed, as seen in Figure 3, in the early stages of training, the CNN layer shows significant differences between clients, but as training progresses, it offers almost no difference. In contrast, the classifier consistently exhibits differences.

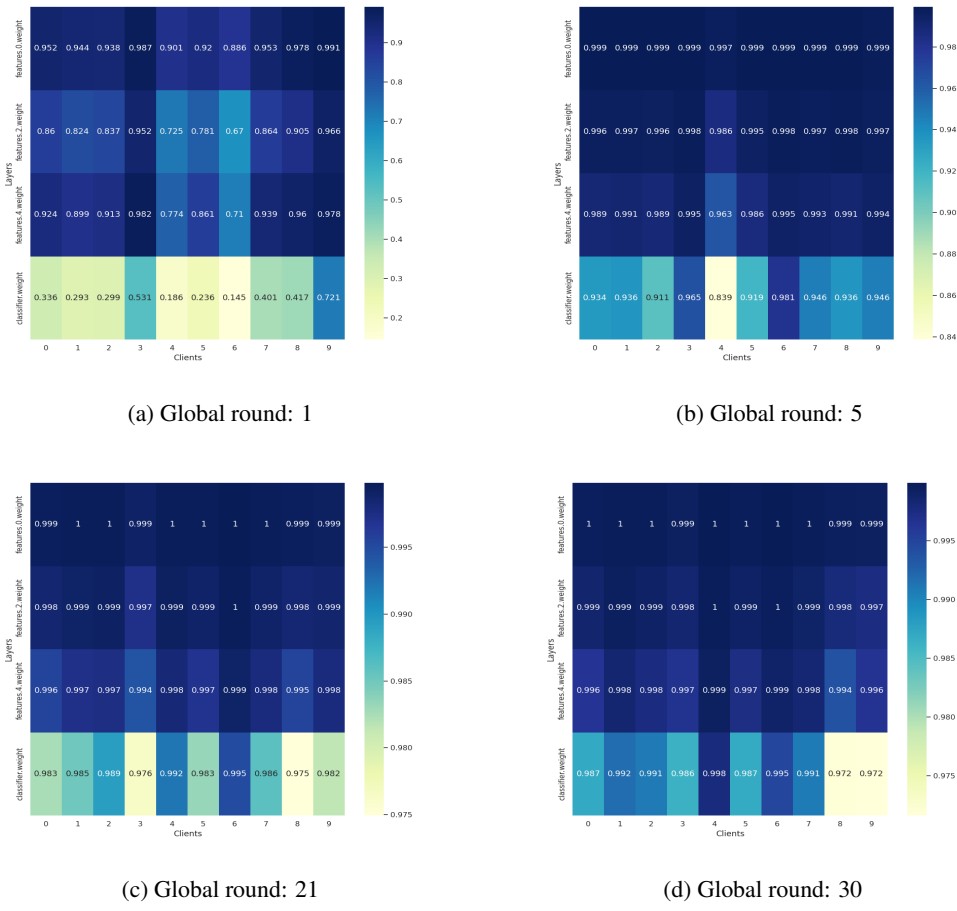

Figure 3: This image represents the cosine similarity between the previous and aggregated global models as the global round progresses.

### C.2   EFFECT OF THRESHOLD $\beta$ IN AGGREGATION BALANCER

Within the aggregation balancer, the threshold $\beta$ serves the dual purpose of clipping to remove outliers and acting as the softmax function's temperature. A more significant value of $\beta$ implies the allowance of more outliers, and including more outliers can lead to a sharper probability distribution. To determine an appropriate value for $\beta$, we trained data with a Dirichlet distribution of 0.1 on CIFAR-10, using ConvNet as a benchmark. Local and global iterations were conducted 10 and 130 times, respectively. The experiment was conducted three times, and when using a $\beta$ value of 3, we obtained results with the highest accuracy and the smallest deviation (see Table 8).

Table 8: Accuracy based on the threshold $\beta$.

| $\beta$ | 1 | 2 | 3 |
|---|---|---|---|
| Accuracy | 62.64 ($\pm$1.2) | 62.65 ($\pm$2.3) | **64.54** ($\pm$0.7) |

### C.3 Observation of distribution $p_i$

Even if the weighted ratio in the global model is identical, the weights of clients enriched with data undergo more updates, leading to substantial changes in magnitude, and thus, are more prominently reflected in the global model.

Figure 4: Histogram of the global model reflection distribution ($p_i$) when simultaneously using both the local balancer and the aggregation balancer. The results are obtained using a Dirichlet alpha of 0.05 on the CIFAR-10 dataset.

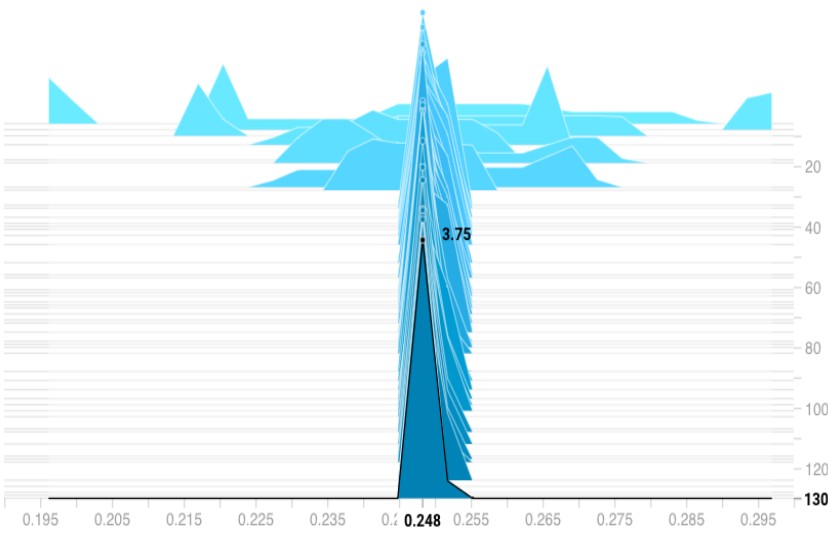

## D Additional Experimental Results

To validate the robustness of the proposed method, we carried out additional experiments using the ResNet-18 model. The experiments were performed under various non-IID data distributions, maintaining the rest of the hyper-parameters consistent with the initial experiments. However, due to the consideration of training time in the case of ResNet, we limited our experimentation to three methods, omitting FedProx and Scaffold.

### D.1 Experiments result on ResNet-18

Table 9: Accuracy comparison with various non-IID settings on ResNet-18 and CIFAR-10 used. The bold text represents the best result.

| Methods | $\alpha = 0.1$ | $\alpha = 0.5$ | $\alpha = 1.0$ |
|---|---|---|---|
| FedAvg | 46.35 | 66.02 | **68.53** |
| FedAvg w/ AB | 39.60 | 63.74 | 67.08 |
| FedNova | 45.10 | **66.19** | 68.29 |
| FedNova w/ AB | 42.45 | 65.50 | 66.84 |
| FedBal (Ours) | **47.03** | 65.98 | 68.41 |
| FedBal w/ AB (Ours) | 42.57 | 65.58 | 67.48 |

