# OpenReview forum: "Knowledge Is Not Wisdom: Weight Balancing Mechanism for Local and Global Training in Federated Learning"
_ICLR.cc/2024/Conference — ICLR 2024 Conference Withdrawn Submission_

### Official Review · Reviewer_9qrP · 2023-10-28

**Soundness:** 2 fair
**Presentation:** 3 good
**Contribution:** 2 fair
**Rating:** 3
**Confidence:** 4

**Summary:**

This papers aims to construct a better global model compared to that of prior works, by adopting (1) a local balancer and (2) an aggregation balancer. The local balancer reduces the client drift by balancing the logit for each class in the client. The aggregation balancer uses the cosine similarity between the global model before local update and the updated model of each client, to measure the importance score of each client. Experimental results how that the proposed scheme performs better than FedAvg, FedProx, FedNova, Scaffold in various federated learning scenarios.

**Strengths:**

1. The paper is in general easy to follow.

2. The proposed scheme performs better than various baselines.

**Weaknesses:**

1. First of all, I feel that the technical novelty of this paper is somewhat lacking. For the local balancer part, FedBal is actually simply adopting existing work from the long-tail / class-imbalanced learning literature. I'm also not very clear with the technical novelty of the aggregation balancer compared with existing works on server-side applied methods.

2. Moreover, the authors are comparing their scheme with only client-side applied methods, even though the authors are proposing both client-side and server-side methods. What is the advantage of FedBal compared with server-side applied methods? Can existing combinations of client-side and server-side methods perform better than FedBal? I believe the authors should address these questions.

3. The authors are considering two datasets, which I believe is not sufficient.

4. Finally, there are several places that makes me feel that the paper is not well-polished or self-contained.
- In section 5.1, the same paragraph is repeating twice.
- In section 5.1, the authors mention that they only focus on cross-silo FL, but also say that 4 out of 20 clients are participating in each round. Maybe the authors are considering cross-device FL?
- How is the outlier balancer $\beta$ determined in the scheme?
- In section 4.1, the authors do not provide details on how $z_j^b$ is actually differs from $z_j$.

**Questions:**

See the weakness above

---

> ### Author Response · Authors · 2023-11-13
>
> Thank you for your feedback. Some of your points may be addressed in my response to another reviewer's question. Long-tail and federated learning have some differences; while long-tail assumes the existence of at least one data point, federated learning allows for the possibility of no data. This difference might have led to questions about the applicability of certain methods in federated learning. However, recent papers from ICCV 2023 on federated learning seem to be proving the effectiveness of similar methods.
>
> Furthermore, most federated learning studies to date have either used FedAvg as is or employed a uniform distribution for aggregation. Server-side methods do exist but often involve extensive computation or use of proxy data, which contradicts the concept of federated learning. We aimed to develop an aggregation scheme that contributes 'fairly' to the global model without additional information. We also wanted to demonstrate that having more data and diverse labels does not always positively impact the global model, a point that may not have been clearly presented in our paper.
>
> Regarding the suggestions to incorporate server-side methods or add datasets, I agree.
>
> Thank you.

---

### Official Review · Reviewer_zEAj · 2023-10-29

**Soundness:** 2 fair
**Presentation:** 3 good
**Contribution:** 2 fair
**Rating:** 5
**Confidence:** 4

**Summary:**

This paper aims to address the data heterogeneity problem in federated learning. The problem is identified as a result of the biased global model. To mitigate these biases, local and aggregation balancers for FL (FedBal) are developed towards specific classes and specific clients, respectively. Experiments have shown the superiority of the proposed balancers when they are combined with existing methods.

**Strengths:**

1. It is good to see a related work summary from the client side and the server side, respectively. This brings about a novel viewpoint to existing FL methods.

2. The proposed aggregation balancer has improved the performance of various FL methods, which demonstrates its effectiveness for different optimization schemes.

3. Experimental settings and training specifications are given in detail.

**Weaknesses:**

1. The core motivation and idea of the local balancer inherit from the method in (Li et al., 2022a). The authors have also mentioned that the loss in Eq. (4) is from (Li et al., 2022a). I think these have limited the novelty and contribution of the proposed method. There is a lack of an explanation of the unique contribution and novelty of the balancer in this paper and how it differs from that in (Li et al., 2022a).

2. Although the proposed method addresses the data heterogeneity problem, it seems that only label-level shift heterogeneity is taken into consideration. As for the feature-level shift heterogeneity, e.g., domain shift, the proposed method seems unable to address it well.

3. From my perspective, more knowledge always leads to a wiser decision, but more data does not lead to more knowledge, considering the data properties can vary a lot in different cases. I think the concepts of knowledge, data, and wisdom can be further clarified.

**Questions:**

None.

---

> ### Author Response · Authors · 2023-11-13
>
> Thank you for your thorough review.
>
> 1. In truth, my intention was to emphasize the effects of the aggregation balancer rather than the performance of the local balancer. However, it seems that this was not adequately conveyed in the paper. Nonetheless, methods like the local balancer have only recently been proven effective in federated learning. I included limited information due to concerns about highlighting effects that are not yet widely recognized.
>
> 2. Federated learning faces challenges in accuracy improvement due to various factors. While separate research on Feature-level heterogeneity is ongoing, to my knowledge, there is not yet a method capable of simultaneously addressing both label-level heterogeneity and feature-level heterogeneity. However, I also believe that both aspects should be considered.
>
> 3. Lastly, I agree that there is a need to clarify terminology and concepts definitively.
>
> Thanks.

---

### Official Review · Reviewer_k5UQ · 2023-10-29

**Soundness:** 1 poor
**Presentation:** 1 poor
**Contribution:** 2 fair
**Rating:** 1
**Confidence:** 5

**Summary:**

The paper proposes a novel FL algorithm to mitigate the bias in model aggregation by assigning higher weights to the marginalized local models. Experimental results show the proposed method can improve the test performance of non-IID FL and combine with the existing methods.

**Strengths:**

The paper attempts to solve an interesting problem, aggregating local models in a more effective way, instead of naively averaging.

**Weaknesses:**

1. The paper isn't well written and well-organized. For example, there are two colonial paragraphs in the section 5.1, begining from 'Our experiments primarily used a 3-layered ConvNet...'.  And the overall training procedure is not illustrated in the paper and there is no a graph to show the workflow.

2. There is no theoretical guarantee in this paper. The proposed method FedBal is tottally empirical.  However, the method  can not outperform all other baselines in the experiments. For example, in table 5, FedBal shows better results than the existing methods in all cases
except $\alpha = 0.1$. It is very strange that FedBal works with higher heterogeneity $\alpha = 0.05$ and lower  heterogeneity $\alpha = 0.5, 1$ but does not work with $\alpha = 0.1$.

3. The experiments are not comprehensive. Three benchmark datasets are sufficient for validating the effect of method. And there are some related prior studies not in the baselines. For example, [1][2], using bayesian theorem to aggregate the local models.

4. The number of clients is too small. only 4 clients paricipating in the training each round. The FL system in the experiments is too small.


Reference:

[1] Yurochkin M, Agarwal M, Ghosh S, et al. Bayesian nonparametric federated learning of neural networks[C]//International conference on machine learning. PMLR, 2019: 7252-7261.

[2] Wang H, Yurochkin M, Sun Y, et al. Federated learning with matched averaging[J]. arXiv preprint arXiv:2002.06440, 2020.

**Questions:**

1. Do all clients have the same number of data? The number of data decides the weight assigned to the local model $p_{i}$. What if a client has larger local dataset but with imbalanced data. Does this client have 'extensive knowledge' or marginalized model?

2. How do you aggregate the models in the server? The paper just shows the objective function of FedBal in equation (9).

---

> ### Author Response · Authors · 2023-11-13
>
> Firstly, I would like to express my gratitude for your comment. The answer to your question is as follows:
>
> 1. The scenario presented refers to a situation where a single client possesses a diverse range of labels and a substantial amount of data, akin to the dir a=0.05 case. If a representative client holds a significant amount of data for all labels, then FedAvg would naturally be advantageous. However, if even one class of data is less represented, methods like FedAvg fail to adequately reflect these lesser-represented classes during the aggregation phase.
>
> 2. In federated learning, aggregation is typically performed at the server level. It was an oversight on my part to represent my method solely through equations, without accompanying figures or algorithmic representations.

---

### Official Review · Reviewer_6uTg · 2023-11-03

**Soundness:** 2 fair
**Presentation:** 2 fair
**Contribution:** 2 fair
**Rating:** 5
**Confidence:** 4

**Summary:**

The paper aims to tackle data heterogeneity in Federated Learning (FL) by introducing Federated Learning Balancer (FedBal). This addresses the bias introduced when servers average a client's model based on data volume or sharing degree. FedBal incorporates two components: the local balancer, regulating preferences in local models, and the aggregation balancer, which calculates cosine similarity to adjust aggregation weights for the global model. It shows an enhancement of 2% to 7% in marginalized clients' models compared to those with abundant data. The experiments demonstrate FedBal can improve global model accuracy by 3% to 4% compared to alternative methods.

**Strengths:**

1. Originality: Empirical evidence indicates that focusing on marginalized clients during aggregation can prevent model overfitting, suggesting a new perspective. The method's simplicity aids reproducibility and further research.

2. Significance: Balancing mechanisms on both client and server sides show potential in enhancing model accuracy, proposing a universal solution to data heterogeneity in Federated Learning.

3. Clarity and Replicability: The paper's structure and method simplicity contribute significantly to its clarity and replicability.

**Weaknesses:**

1. Editorial Imperfections: Repetitive statements and editorial issues affect the paper's readability and require careful revision.

2. Limited Novelty: The novelty of the local balancer and aggregation equalizer is constrained. They draw heavily from prior research, limiting their originality.

3. Inconclusive Experiments: The experiment lacks comprehensive contemporary baseline comparisons, limiting the persuasiveness of its findings.

**Questions:**

1. Figure 1 mentions that in the case of a uniform segmentation of labels, labels that are shared more among customers (0, 1, 6, 7, etc.) usually show a higher accuracy rate. However, some labels, notably label 6, experience a decrease in accuracy, while label 7 maintains a consistent accuracy. Moreover, certain labels (2, 3, 4, 8, 9) also display decreased accuracy, with only labels 0 and 1 showing improvements. These results might not unequivocally support the assertion that FedAvg yields subpar performance.

2. Table 1 presents accuracy data where the Without exp (Original) configuration yields higher accuracy. The interpretation of these results is somewhat perplexing. Does this imply that the absence of the aggregation balancer results in superior performance?
3.	Analyzing Table 2, it is observed that, on the CIFAR10 dataset, as data heterogeneity increases, the local balancer's performance falls behind the baseline. Could the reasons for this performance differential be elucidated?
4.	In Table 4, a notable deterioration in performance is observed when α is 0.1 and the aggregation balancer is employed. Could you provide an explanation for the specific circumstances in which a decrease in performance occurs at certain levels of data heterogeneity?
5.	Could a dedicated ablation study, focusing solely on the use of the aggregation balancer, be provided to validate its efficacy?
6.	The proposed method appears to lack consistency. In some instances, the local balancer may underperform compared to the baseline in specific settings, while the aggregation balancer can also degrade performance in certain scenarios. Paradoxically, the combined use of both balancers in certain settings results in a marked improvement in accuracy, like in CIFAR10 and α is 0.05. Could the underlying reasons for these variations in performance be explained?

---

> ### Author Response · Authors · 2023-11-13
>
> Thank you for your comment. It seems that my explanation was not clear enough, leading to numerous misunderstandings. Additionally, I acknowledge that our analysis of the experiment was insufficient. The response to your question is as follows:
>
> 1. There appears to be a misunderstanding regarding the figure. The results depicted here are not from the application of the method we propose, but rather, they describe outcomes commonly observed in federated learning using aggregation techniques. The intention was not to suggest that uniform division is more suitable than FedAvg. Instead, our aim was to indicate the occurrence of a trade-off relationship.
>
> 2. I acknowledge the ambiguity in the expression of 'with/without exp' in the table. Our method was intended to highlight the differences when using values obtained through Cosine similarity either in a proportional expectation or inversely. Both methods presented here are experiments applying the Aggregation Balancer. Cosine similarity values are closer to 0 when less similar, and closer to 1 when more similar. (Though they can be as low as -1, negatively similar values indicate conflicts in aggregation, which we consider as being less similar.) Therefore, directly applying the exponential function to the cosine similarity values implies placing greater importance on the global model for clients similar to the global model (i.e., clients with less or unique data, or those not significantly trained). Conversely, using an inversely exponential function implies giving more weight to clients that have undergone significant changes (clients with more data).
> Our findings suggest that, in the stage of aggregating within the Global model, it may be beneficial to prevent overfitting by assigning a higher weighting ratio to clients that have undergone less change.
>
> 3. Federated learning differs from the long-tail problem in that each client may not possess any data at all. The long-tail issue assumes at least some data is held, which limits its applicability in situations like federated learning where not all data is necessarily available.
>
> 4. In fact, we also observed this phenomenon and conducted several experiments. We found that in almost all situations, our method was not effective when a=0.1. We speculate that this may be due to a conflict arising from the trade-off relationship between the local balancer and the aggregation balancer. While the aggregation balancer shows positive effects in non-iid environments, it becomes increasingly similar to FedAvg as the data approaches IID. The local balancer operates in the exact opposite manner. Therefore, it appears that ultimately the advantages of each are offset by the other.
>
> 5. We believe that our experiments applying Global aggregation to FedAvg have demonstrated its effectiveness, as evidenced in Table 3 and Table 4.
>
> 6. Federated learning always faces a trade-off dilemma, which is where our problem definition began. If we follow the majority opinion of those holding abundant information, minority labels get overlooked. Conversely, providing equal opportunities increases the accuracy for minority labels, but leads to information loss in the global model due to the relatively reduced opportunities for clients with more information.
> As a compromise in this situation, our approach involves local balancers optimizing their models as much as possible before transmitting to the aggregation server. This process encourages the global model to reflect minority opinions more significantly. However, ultimately, in the later stages of learning, clients holding more information (i.e., those with more labels) end up contributing more significantly to the global model, thereby enhancing overall learning accuracy.